# Evidence of an active volcanic heat source beneath the Pine Island Glacier

Brice Loose[1], Alberto C. Naveira Garabato[2], Peter Schlosser[3,4], William J. Jenkins[5], David Vaughan[6] & Karen J. Heywood [7]

Tectonic landforms reveal that the West Antarctic Ice Sheet (WAIS) lies atop a major volcanic rift system. However, identifying subglacial volcanism is challenging. Here we show geochemical evidence of a volcanic heat source upstream of the fast-melting Pine Island Ice Shelf, documented by seawater helium isotope ratios at the front of the Ice Shelf cavity. The localization of mantle helium to glacial meltwater reveals that volcanic heat induces melt beneath the grounded glacier and feeds the subglacial hydrological network crossing the grounding line. The observed transport of mantle helium out of the Ice Shelf cavity indicates that volcanic heat is supplied to the grounded glacier at a rate of ~ 2500 ± 1700 MW, which is ca. half as large as the active Grimsvötn volcano on Iceland. Our finding of a substantial volcanic heat source beneath a major WAIS glacier highlights the need to understand sub-glacial volcanism, its hydrologic interaction with the marine margins, and its potential role in the future stability of the WAIS.

[1] Graduate School of Oceanography, University of Rhode Island, Narragansett, Rhode Island, USA. [2] National Oceanography Centre, University of Southampton, Southampton, UK. [3] Arizona State University, School of Earth and Space Exploration, Tempe, AZ, USA. [4] Lamont-Doherty Earth Observatory, Columbia University, New York, NY, USA. [5] Marine Chemistry and Geochemistry, Woods Hole Oceanographic Institution, Woods Hole, MA, USA. [6] British Antarctic Survey, Cambridge, UK. [7] University of East Anglia, Norwich, UK. Correspondence and requests for materials should be addressed to B.L. (email: bloose@uri.edu)

The stability of Pine Island Ice Shelf and the Pine Island Glacier are of paramount importance to sea level rise and the mass balance of the West Antarctic Ice Sheet (WAIS)[1]. Geothermal heat sources and the production of subglacial water can influence the bottom boundary condition that partly determines the glacial mass balance[2–4]. Variability in the subglacial water supply[5], including that caused by intermittent heat flux[6], can lead to ice sheet instability. Thus, the existence of subglacial volcanism impacts both the stable and unstable dynamics of an ice sheet such as the WAIS.

Determining the distribution of geothermal heat flow to the WAIS is complicated by the presence of an extensional volcanic rift system that stretches across Marie Byrd Land from the Pine Island Glacier to the Ross Ice Shelf and into the Ross Sea[7,8]. This is known as the West Antarctic Rift System (WARS). To date, as many as 138 volcanoes have been identified throughout West Antarctica[9], including the presently active Mt. Erebus[10] along the Terror Rift, as well as Mt. Siple[10] and Mt. Waesche[11], which both show evidence of recent activity. However, the locations and extent of volcanic activity along the WARS are debated, because many of these 138 known volcano-like features are buried beneath several kilometers of ice, and some evidence suggests that much of the interior subglacial WARS is dormant[12,13]. Yet, recent direct measurement of the thermal gradient beneath the Whillans Ice Stream have revealed heat fluxes that exceed the background geothermal gradient[4]. The apparent surface deformations in the WAIS thickness also suggest localized heat fluxes that are most likely volcanic due to their intensity[14,15], while ash layers from ice cores reveal more recent eruptions[16]. Last, the detection of earthquakes as recently as 2010 suggest magma migration beneath the Executive Committee mountains, in a region of Marie Byrd Land where seismic studies have revealed thin crust and low-density mantle material beneath[13]. Despite the accumulation of evidence, definitive proof of contemporary subglacial volcanism in West Antarctica is still missing.

Subglacial volcanism implies melt and subglacial water has been observed through active seismics[17,18]. However, subglacial hydrology can be driven by non-volcanic geothermal heat and friction between the bedrock and the ice sheet, and to date there is no direct evidence of melt by present day volcanism beneath the WAIS. Consequently, the magnitude of subglacial meltwater production and transport remains unknown. Here we report on helium isotope and noble gas measurements that provide geochemical evidence of subglacial heat flux that can only be volcanic in origin and of subglacial meltwater production that is subsequently transported into the cavity of the Pine Island Ice Shelf.

Presently, the greatest contributor to ice shelf instability around Antarctica appears to be an increase in ocean heat supply to the cavities of Antarctic ice shelves[19]. Circumpolar Deep Water (CDW) is the primary heat source for melting glacial ice and its increased presence on the Amundsen Sea continental shelf has been implicated in the rapid melting and grounding line retreat observed beneath the Pine Island Glacier[19–21] and in the atmospheric warming along the western Antarctic Peninsula[22]. The ocean–atmosphere mechanisms that draw more CDW onto Antarctic continental shelves are challenging to characterize and remain poorly understood[23], although the concentration and distribution of CDW and its year-to-year variations have revealed connections to climatic changes in the regional winds[21,24].

In addition to temperature and salinity, helium isotopes are commonly used as a tracer for CDW around Antarctica, because CDW is typically the only source of elevated 3He in Antarctic coastal waters[25]. Therefore, we first describe the isotopic background against which the evidence of volcanism can be contrasted. The $^3$He/$^4$He isotope ratio is typically expressed in percent (%) deviation from the atmospheric ratio ($R_A$) as $\delta^3$He =

$(R_{obs}/R_A - 1) \times 100$ at abundances typically found in the ocean. Details of CDW geochemistry can be found in the Supplementary Note 1. Six expeditions to the Pacific and Atlantic sectors of the Southern Ocean provide 1610 $\delta^3$He measurements (Fig. 1). These historical data show maximum values of $\delta^3$He in the core of CDW in the Weddell Sea (Atlantic Sector) of 10.2%[26], and in the Ross and Amundsen Seas (Pacific Sector) of 10.9%, all of which can be traced to subpolar mid-ocean ridge systems in the Pacific[27], Indian and Atlantic Oceans[28].

CDW is modified through ventilation on the continental shelves and this reduces the $\delta^3$He in continental shelf waters. In the Amundsen Sea, CDW can penetrate along troughs to reach ice shelves[29] at potential temperatures ($\theta$) that range from $\theta = 0.5$ to 1.2 °C with salinities $S > 34.6$[21]. This variable modification of CDW is also reflected in the $\delta^3$He from the Amundsen Sea: the warmest water in Pine Island Bay (PIB, blue box in Fig. 1) in 2007 exhibited $\theta = 1.24$ °C and $\delta^3$He = 9.79%. In 2014, the warmest water in PIB was characterized by $\theta = 1.14$ °C and $\delta^3$He = 9.1%. This water is found in the deep troughs of the continental shelf between 600 and 1000 m. However, the two expeditions to PIB in 2007 and 2014 have revealed seawater exhibiting $\delta^3$He values that reach a maximum of $\delta^3$He = 12.3%, which stands well above the deep $\delta^3$He maxima in CDW (Fig. 2a). This excess $\delta^3$He is most prominent at the Pine Island Ice Shelf front (Fig. 3), and thus far was not encountered further west in the Amundsen Sea, nor at the front of adjacent ice shelves, and neither in the Ross[30] nor the Weddell Seas[31], including the Ross and Filchner-Ronne ice shelves. The anomalously high $\delta^3$He values in PIB also coincide with elevated neon concentrations (colored circles in Fig. 2). Neon concentrations above atmospheric equilibrium are found within melted glacial ice[25,32], suggesting that the excess $^3$He is associated with glacial meltwater at the front of the ice shelf. Significantly, the excess $^3$He is not distributed evenly and is not found near the strongest meltwater outflow[33]. This suggests that the excess $^3$He signal originates in a unique, localized meltwater source, rather than a diffuse distribution that is found in all meltwater along the cavity front.

## Results

**Testing for a unique $^3$He source.** To establish whether the $^3$He distribution observed in PIB can be explained by mixing between CDW and the other Amundsen Sea water masses, we employed a linear mixing model—Optimal Multiparameter analysis (OMP)—to map the range of likely $\delta^3$He values. The principal water masses in PIB can be categorized as modified CDW[34], Amundsen Surface Water (ASW), and Glacial Meltwater (GMW). CDW is the densest water mass in the Amundsen Sea and dominates the water column below 400 m and the density horizon of $\sigma_\theta = 27.89$ kg m$^{-3}$. We define ASW as water found between the ocean surface and the mixed layer base, with $\theta$ values near the seawater freezing point ($-1.9 < \theta < -1.8$ °C), salinities ranging from 33 to 34.2, and $\delta^3$He values that range from the atmospheric equilibrium value of $\delta^3$He = $-1.7$ up to $\delta^3$He = 2.5%. The hydrographic properties of ASW reflect the fact that in certain regions/times the ocean surface equilibrates with the atmosphere, but can also show a strong disequilibrium as a result of extensive sea ice cover. $\delta^3$He in pure GMW should be close to zero, as it is derived from the air trapped in glacial ice. This range of variation defines the mixing space between warm, salty CDW, and colder, fresher air-equilibrated water, or between CDW and water from the previous winter.

The samples obtained in PIB in 2007 and 2014 do not follow the mixing space mapped out by the CDW–ASW water masses (Fig. 2). A fork in the $\delta^3$He distribution occurred between $\theta = -1.5$ and $-0.5$ °C, with $\delta^3$He exceeding the average CDW end

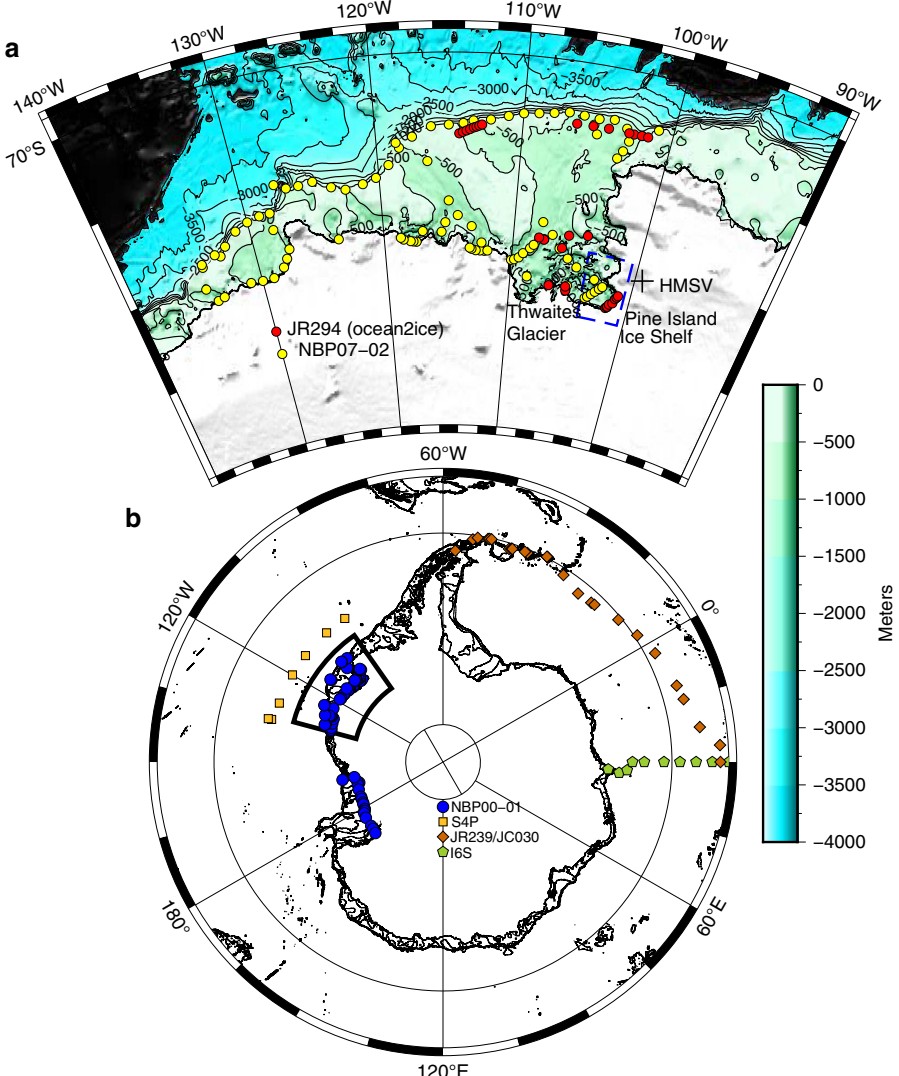

**Fig. 1** Map of $^3$He measurements around Antarctica. Station locations for helium/neon data used in Fig. 2. **a** An expanded view of the Amundsen Sea and locations of helium/neon hydrographic stations during NBP07–02 (2007, yellow) and JR294 (2014, red). The " + " demarcates the approximate location of the Hudson Mountain Subglacial Volcanoes. The blue dashed line demarcates Pine Island Bay. The mapped region of the Amundsen Sea is indicated by the inset box in **b**, which depicts the regional and offshore helium isotope hydrography, also included in Fig. 2

member (9.15 ± 0.65) in 2014 by up to 33% ($\delta^3$He = 12.2) and in 2007 by 35% ($\delta^3$He = 12.3). The $\delta^3$He in excess of the CDW maximum was found above 300 m and primarily at the front of the Pine Island Ice Shelf (Fig. 3).

Altogether, 28 of the 106 $\delta^3$He samples measured in PIB exceeded the upper limit of the 99% confidence criterion (see Methods and Supplementary Figure 3) for $\delta^3$He produced by mixing between CDW and ASW, strongly suggesting that there is another source of $^3$He in PIB in addition to CDW.

**Identifying the possible sources of local $^3$He production.** The mantle is the largest reservoir on the planet, but $^3$He is also produced via $^3$H decay in the atmosphere and during detonation of nuclear devices[35], although very little thermonuclear $^3$H was deposited in the Southern Ocean[36]. The maximum measured $^3$H in the Amundsen sea during 2014 was 0.13 TU, which corresponds to $1.4 \times 10^{-17}$ moles $^3$He kg$^{-1}$[37]. For comparison purposes, $\delta^3$He = 1 corresponds to roughly $1.3 \times 10^{-15}$ moles $^3$He kg$^{-1}$, or a factor of 100 greater than the tritiugenic $^3$He. In other words, the presence of $^3$H can account for at most 0.2% of the $^3$He excess

that was observed. The balance of production with a 12.43-year half-life and air–sea gas exchange means that the actual tritiogenic $^3$He would be even less. In summary, the $^3$He contribution from tritium decay is insignificant.

Seismic, magnetic, and gravity swaths from the Amundsen Sea indicate the existence of thinning crustal features running NE to SW between 72 and 74 °S. In this region, the distance to the Mohorovičić discontinuity is thought to be 22–24 km below the earth's surface[38]. However, these features are north of PIB and have not been associated with crustal motion since before 90 Ma ago. The excess $^3$He found in PIB occurred primarily at the front of the ice shelf cavity and above 500 m depth, indicating that if the thin crust were the source of excess helium, we would observe its trace in the deep waters of the Bay before it could mix into the lighter meltwater at the surface and front of the ice shelf.

The existence of a tectonic fissure directly beneath the Pine Island Ice Shelf might also be a source for the mantle $^3$He observed at the front. However, such a feature would also produce a strong thermal anomaly that was not consumed by melting ice. This anomaly would likely disturb the thermohaline structure of

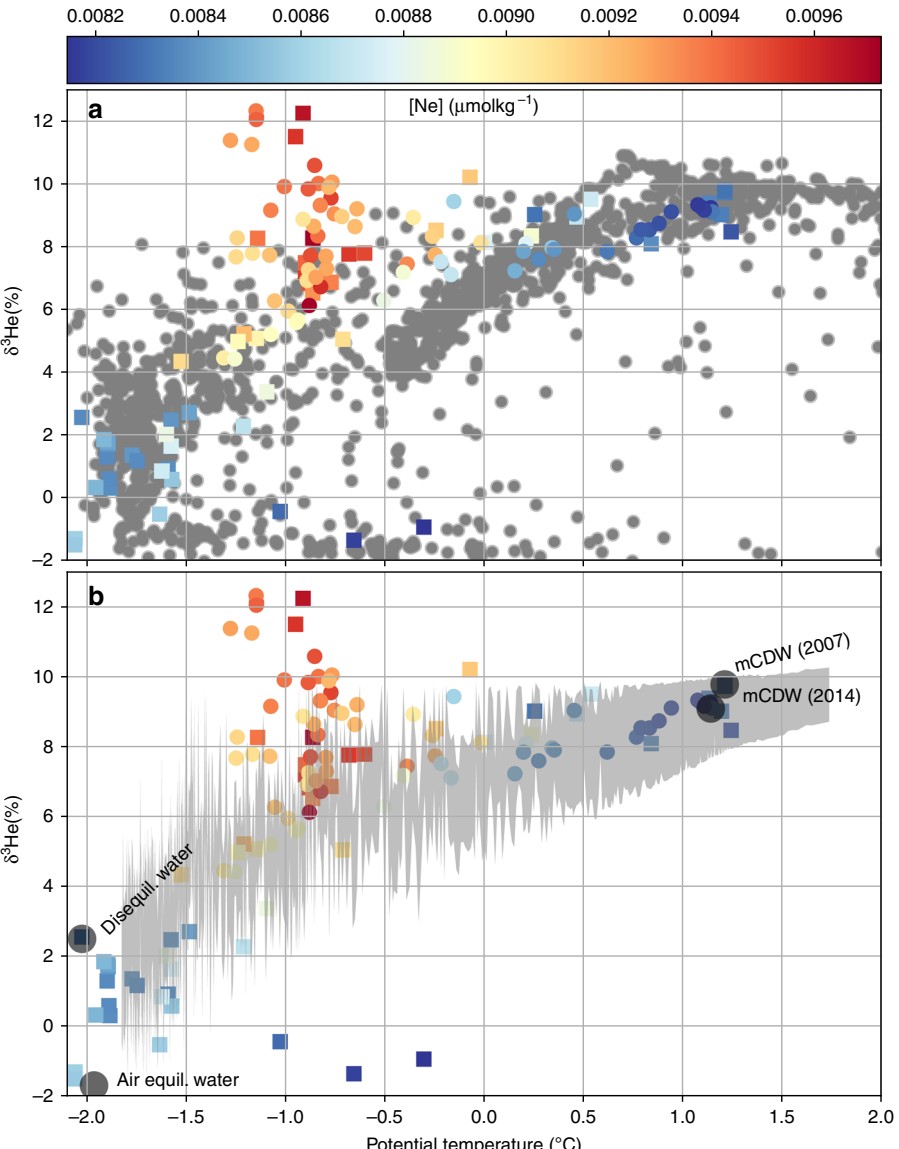

**Fig. 2** Distribution of $^3$He vs. potential temperature. **a** The gray circles are values of dissolved $\delta^3$He vs. seawater potential temperature (°C) from six expeditions to the Pacific and Atlantic sectors of the Southern Ocean: NBP00-01, NBP07-02, JR239, I6S, S4P, and JR294 (Ocean2ice), $N = 1610$. The samples colored by neon concentration are the 106 gas samples collected in Pine Island Bay during NBP07-02 (squares) and JR294 (circles). **b** The gray shaded area represents the 99% confidence region using a bootstrap resampling statistic to reproduce the observed values of $\delta^3$He from the water mass mixing model that has been constrained by neon, $\theta$ and $S$. In total, 28 of the 106 helium isotope samples exceed the upper confidence limit

the ice shelf cavity and appear as a mismatch in cavity heat budget calculations[39]. The Autosub mapping expeditions into the Pine Island ice cavity have not revealed thermal anomalies of this nature[19].

The $^3$He/$^4$He isotope ratio that is used to compute $\delta^3$He can also be affected by the production of $^4$He through the radioactive decay series that begins with $^{238}$U, naturally abundant in many rock types within the continental crust, which can subsequently leach into groundwater and sediment porewater[40]. The $\delta^3$He signal that we observe at the front of the Pine Island Ice Shelf may include additional $^4$He from crustal rocks, but this incorporation drives the $^3$He/$^4$He isotope ratio toward low values, which is the opposite direction from that of the mantle[41], so additional $^4$He production would mask or underestimate the mantle helium component. There are no known processes for removing $^4$He gas, save bubble formation, or diffusive degassing, which would affect all the dissolved gases in a similar manner.

Lacking a heat source beneath the cavity or in PIB, the next most likely source is upstream of the cavity beneath or within the ice sheet. The observation of debris-rich basal layers in icebergs at the grounding line reveals the transport of glacial till and rubble across the grounding line. These debris-laden glaciers are not a likely $^3$He source. Mantle $^3$He escapes during magma degassing, which produces steam and volatile gas transport in adjacent hydrothermal fluids[42]. Even if the glacial debris is rich in basalts, these cooled magmas have already lost much of their $^3$He burden during the cooling process. Hereafter, we refer to the magma-driven hydrothermal heat transport as the "volcanic heat flux."

**Implications of excess $^3$He.** If the mantle helium source is located beneath the Pine Island Glacier or its tributaries, these geochemical measurements, collected at the front of the ice shelf cavity, reveal a subglacial hydrologic flow path that exchanges

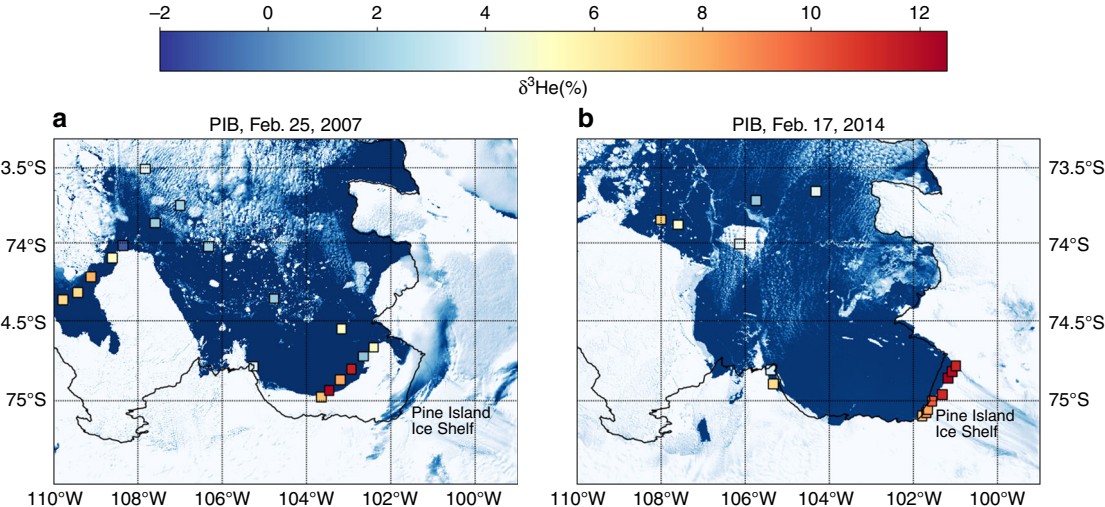

**Fig. 3** Map of elevated $^3$He samples from 2007 and 2014 in Pine Island Bay. MODIS images of Pine Island Bay on **a** 25 February 2007 and **b** 17 February 2014. In 2007, fast ice precluded sampling directly at the front of the ice shelf cavity. The colored squares depict the maximum value of $\delta^3$He in the top 300 m of the water column

water with the marine-terminating margin of the glacier, and that volcanic heat may be contributing to subglacial melt beneath the Pine Island Glacier. Radar data show that ice sheets heave under tidal influence[43], suggesting that water could be exchanged past the ice shelf grounding line. Stable isotopes from sediments beneath the Whillans Ice Stream also indicate a small percentage of seawater intrusion[44]. Whereas these are apparently the first geochemical measurements from the Amundsen Sea demonstrating the transport of sub-basal meltwater to adjacent coastal seas, this process is well-documented in subterranean groundwater discharge[45] and there is evidence of similar discharge beneath the Ross Ice Shelf, although helium isotopes suggest that at this location the subglacial water interacted mainly with continental crust, rather than volcanic rocks[46].

Considering the abundance of volcano-like features along the WARS[9], ice sheet contact with a volcanic heat source is the mostly likely source of excess $^3$He. Volcanism in the WARS was most active around 30 Ma before present[47], but there is evidence of more recent eruptions[48]. The adjacent Thwaites glacier, which drains to the Amundsen Sea, shows strong radar returns that indicate subglacial meltwater, suggesting volcanism and high localized heat flux[8,15]. However, the Thwaites drainage is isolated from the Pine Island drainage, so meltwater from the Thwaites is not a likely source for the mantle helium we observed. Instead, the Pine Island ice stream funnels through a deeply scoured subglacial trough[49] that receives ice from tributaries to the east in the Hudson Mountain range. Rocks from the exposed portion of the Hudson Mountains, including Mt. Manthe date between 4.6 and 5.0 Ma before present, with some observations of presently active fumaroles[48]. Within the Hudson Mountains, this network of between 3[10] and 11[9] volcanic landforms lie upstream of glacial Tributary 6 and of PIB. Evidence of subglacial volcanism is present in the form of an ash deposit covering some 23,000 km$^2$ near 500 m depth in the ice sheet[48]. This ash layer reveals an eruption that dates to approximately $2.22 \pm 0.240$ ka before present[48]. These subglacial volcanoes in the Hudson Mountain range (Fig. 1) are the most plausible source of mantle helium to the Pine Island subglacial drainage network.

**Calculation of volcanic heat flux**. The excess neon found in samples with excess $^3$He reveals a connection between mantle helium and glacial meltwater production, which is consistent

with the production of subglacial melt by volcanic heat beneath the grounded Pine Island Glacier. We have estimated this volcanic heat content using the average of 17 reported estimates of $^3$He/heat ratio (HR) from subsea hydrothermal vents. Lupton et al.[50] provide a summary of the HR values, whereas Jenkins et al.[37] give a recent estimate for the Atlantic spreading center. The mean and standard deviation of the literature values from subsea floor vents yield a $^3$He/HR $= 17 \pm 6 \times 10^{16}$ J per mol $^3$He (Supplementary Table 2). We computed the $^3$He excess ($^3$He$_{exc}$) as the difference between the measured $^3$He and the $^3$He predicted by the linear mixing model ($^3$He$_{OMP}$, see Methods). The $^3$He$_{exc}$, expressed in mol kg$^{-1}$ of seawater divided by HR provides an estimate of volcanic heat content in Joules per kilogram of seawater (J kg$^{-1}$).

Based on the observed $^3$He excesses, the mantle-derived heat at the front of the ice shelf cavity is $32 \pm 12$ J kg$^{-1}$ of seawater. This excess heat is small compared to the heat content of CDW[20] (ca. 12 kJ kg$^{-1}$), demonstrating that volcanic heat does not contribute significantly to the glacial melt observed in the ocean at the front of the ice shelf. This interpretation is consistent with our understanding of melt dynamics beneath the Pine Island Ice Shelf - that most of the basal melt occurs within the cavity, as a result of ocean heat supply[20]. Yet, the relatively dilute volcanic heat source may be much more concentrated at the time of contact with the ice sheet, and the magnitude more significant when compared to the background geothermal heat supply to the grounded glacier. We infer the heat flux to the ice sheet using observations of the cavity circulation at the ice shelf front (Eq. 3).

After accounting for uncertainty in the gridded interpolation of $^3$He data, temporal variability in strength of perpendicular velocity, and uncertainty in the estimate of HR (see Methods), the volcanic heat flux exiting the cavity beneath the Pine Island Ice Shelf was $Q = 2500 \pm 1700$ MW in 2014, with peak flux occurring between 50 and 250 m below the ocean surface (Fig. 4). How does this heat source compare with present day active volcanoes, hotspots or hydrothermal vents? Heat energy released by volcanoes and hydrothermal vents suggests that the heat source beneath the Pine Island Glacier is roughly 25 times greater than the bulk heat flux from an individual dormant volcano. A survey of 51 dormant or quiescent volcanoes indicates that they release less than 500 MW of heat energy, with an average of 97 MW[51,52]. These dormant volcanoes release the majority of their

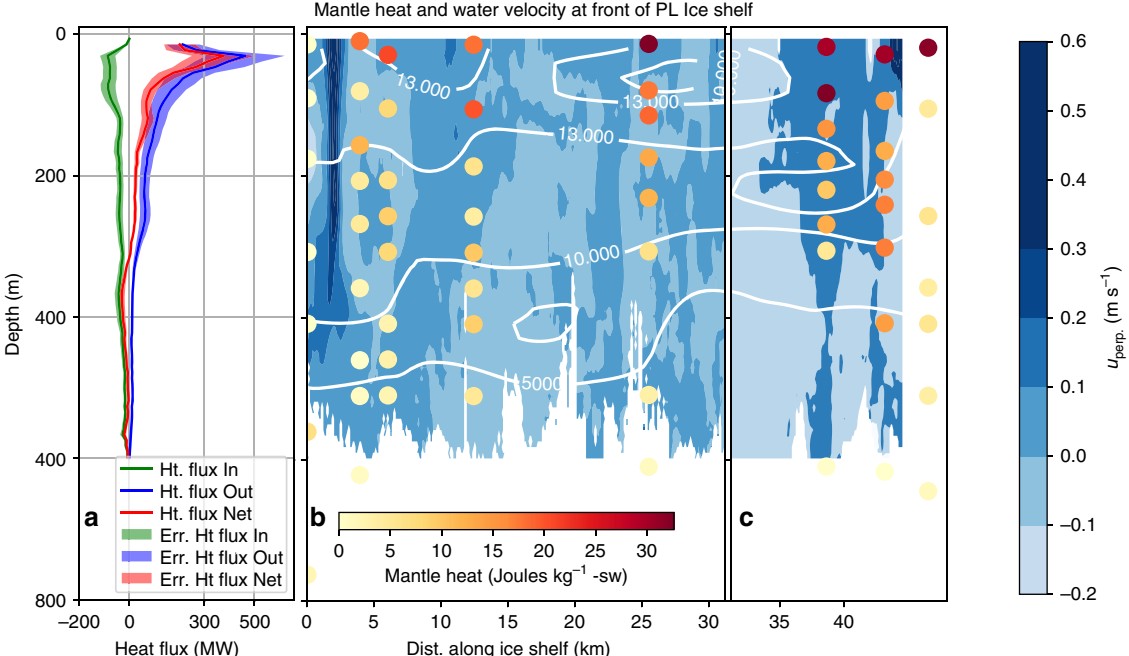

**Fig. 4** Distribution of velocity and mantle heat at the front of Pine Island Ice Shelf. Section along the edge of the Pine Island Ice Shelf in 2014 (as indicated by colored squares in Fig. 3b). **a** Shows the discrete sum of Inward, Outward, and Net mantle heat flux in Megawatts (MW) at each depth horizon in the heat flux grid. **b, c** Depict water velocity perpendicular to the front of the Pine Island Ice Shelf ($u_{perp}$, gridded data) as filled contours; positive values indicate flow out of the ice shelf. The break at 32 km exists because the two sections were collected at different times, separated by less than 24 h. Colored circles depict the discrete estimates of mantle-derived heat in Joules per kg of seawater, determined from the excess $^3$He. The white contour lines are estimates of glacial meltwater (‰) from the linear mixing model

heat into their crater lakes (50–250 MW), but fumaroles and geysers may intermittently contribute an additional 50 and 1000 MW[42]. See Supplementary Table 3 for a review of the heat estimation methods.

The heat flux liberated from an active volcano is considerably greater: measurements collected over the past four decades show that Grimsvötn, one of Iceland's most active volcanoes, releases 4250 MW[53] through its crater and into the ice fields along its slope. A similar measurement taken from Nyiragongo volcano in the Democratic Republic of Congo, revealed that magma convection before the 1977 eruption released at least 16,000 MW of heat energy[54].

Whereas dormant volcanoes release hundreds of MW of heat, submarine vent fields along active mid-ocean ridges can release thousands of MW or more. The Southern Symmetrical Segment and the Endeavor Segment of the Juan de Fuca Ridge produce heat fluxes of 1700 and 580 MW, respectively[55]. The Lucky Strike Field along the East Pacific Rise produces 3800 MW of heat energy through smokers and hydrothermal vents[56].

It is worth noting that the volcanic heat flux reflected by excess $^3$He only captures convective heat transfer via hydrothermal fluids. The $^3$He tracer does not capture sensible and conductive heat transfer, which can also be elevated as a consequence of thin crust and a proximal magma heat source[57]. Consequently, 2500 MW may be an underestimate of the total volcanic heat supply.

**Implications of a volcanic heat source**. The impact of the inferred volcanic heat flux on the flow characteristics of Pine Island Glacier depends upon the intensity of the volcanic heat flux (heat flux per unit area) at the base of the ice sheet and possibly upon the temporal variations in this heat source, because transients in the subglacial melt supply have the greatest impact on the ice sheet sliding rate[5]. We lack the information needed to estimate the heat flux intensity with present data sources,

but we can compare it with other natural systems. The heat flux intensity from submarine vents in the Gulf of California is 1900 mW m$^{-2}$ on average, and 15,000 mW m$^{-2}$ at a maximum, but such intense fluxes would likely manifest as large deviations in the surface of the Pine Island ice sheet elevation.

A recent set of model experiments that emplaced a mantle plume at various regions beneath the WAIS revealed that heat flux greater than 150 mW m$^{-2}$ leads to high melt production and subglacial drainage events[58]. The experiments used mantle plumes that varied from 50 to 300 km in radius; if the 2500 MW of volcanic heat beneath Pine Island Glacier originated from a plume in this size range, it would imply a glacial heat flux of between 318 and 9 mW m$^{-2}$. If the plume were large (i.e. 300 km radius), the heat flux would be well below the canonical background of 50–70 mW m$^{-2}$; conversely, a 50 km mantle plume would suggest intensive subglacial heating and likely melt. These model experiments did not include the Pine Island Glacier within their domain, but we note that the adjacent Thwaites Glacier proved largely insensitive to the presence of a mantle heat source. Basal friction is high beneath the Thwaites Glacier leading to significant basal heat production and the additional heat from a mantle plume did not drastically alter the ice stream velocity[58].

**Distribution of volcanic $^3$He between 2014 and 2007**. In 2014, the volcanic $^3$He excess was concentrated within a meltwater outflow located across the eastern and central sections of the front of the ice shelf, but did not appear in the strongest meltwater outflow, which occurs at the western end of the section[33]. In 2007, the $^3$He excess was not as broadly distributed at the ice shelf front, and the most excessive values were found further to the west (Fig. 3). The difference in the location and distribution of the excess $^3$He has several possible explanations. One is a change in the strength of the $^3$He source, and therefore a change in volcanic heat flux between the 2 years. The difference might also reflect a

change in the subglacial hydrology that delivers ${}^{3}$He to the ice shelf cavity. The ice shelf underwent rapid and extensive grounding line retreat between 2007 and 2014[19], which may have altered the hydrostatic pressure gradient driving subglacial flow across the grounding line[59]. Alternatively, the grounding line retreat may have produced a change in the cavity circulation and entrainment of subglacial meltwater.

One additional complication to the comparison of 2007 and 2014 measurements is the barrier of fast ice that kept NBP07-02 from reaching the front of the ice shelf, resulting in a transect of water samples about 50 km away from the ice shelf, further out in PIB during 2007. The difference in location of the excess ${}^{3}$He may have been a result of the anti-cyclonic circulation in the Bay, which predictably advects the excess ${}^{3}$He toward the west upon exiting the cavity[60].

## Discussion

The mantle ${}^{3}$He observed at the front of the Pine Island Ice Shelf, first in 2007 and again in 2014, reveals the presence of a volcanic heat source upstream of the Ice Shelf. The observation of this unique helium isotope signature, together with what is known of the bed forms and fluvial morphology of the Glacier suggests that this volcanic heat source lies within the Hudson Mountain range, and is driving a subglacial melt that subsequently crosses the ice shelf grounding line. Our calculations indicate that the volcanic heat source is comparable in magnitude to the active vent fields found along ocean spreading centers. The inferred heat supply is more than ten times the heat energy released by dormant (but not extinct) shield volcanoes on land.

These geochemical measurements provide an independent line of evidence of present day subglacial volcanism in Marie Byrd Land. They also support a growing list of studies revealing that regional volcanism is a recurring characteristic of the basal boundary beneath the WAIS. The present estimate of convective volcanic heat flux alone suggests a heat source of $Q = 2500$ MW, which is ~ 50% as large as the Grimsvötn volcano on Iceland, even before sensible and conductive heat flux have been accounted for. Simulations of the adjacent Thwaites Glacier may suggest that such a heat source will not significantly alter the subglacial melt rate in comparison with the high rate of friction[58], but this could be circular argument if volcanic heat supply is already part of the recipe of processes leading to high velocity and frictional heating of the ice streams in the Pine Island and Thwaites Glacier. The magnitude and the variations in the rate of volcanic heat supplied to the Pine Island Glacier, either by internal magma migration[8], or by an increase in volcanism as a consequence of ice sheet thinning[61], may impact the future dynamics of the Pine Island Glacier, during the contemporary period of climate-driven glacial retreat.

## Methods

**OMP calculation**. The water mass tracers used to constrain the OMP solution were potential temperature (°C), salinity (no units), and neon concentration (μmol kg${}^{-1}$). The concentration of ${}^{4}$He was not used, because of the potential covariance between ${}^{3}$He and ${}^{4}$He. Helium-4 can increase as a consequence of uranium decay in the continental crust[41], which could mingle with the ${}^{3}$He signal as a result of passing through sedimentary pore spaces beneath the glacier.

The OMP uses a non-negative least squares method to resolve the relative contributions of three water masses—CDW, ASW, and GMW,

$$\mathbf{f} = \text{inv}(\mathbf{C}^{T}\mathbf{w}\mathbf{C})\mathbf{C}^{T}\mathbf{w}\mathbf{y}, \ \mathbf{f} \geq 0. \tag{1}$$

where $\mathbf{C}$ is the matrix of tracer properties in CDW, ASW, and GMW, $\mathbf{w}$ is a weight matrix, and $\mathbf{y}$ is the vector of observed tracer concentrations in each water sample. The weights in $\mathbf{w}$ are diag (0.5,0.03,125) for temperature, salinity, and neon, and were determined such that each element in $\mathbf{C}^{T}\mathbf{w}$ has an order − 1 magnitude to ensure that each water mass tracer exerts a proportionate influence on the solution. The result, $\mathbf{f}$, contains the relative fraction of each water mass in the given sample, which was used to reconstruct the ${}^{3}$He value ($\delta^{3}\text{He}_{\text{OMP}}$) that would be expected

from mixing between water masses in the Amundsen Sea,

$$\delta^{3}\text{He}_{\text{OMP}} = \delta^{3}\text{He}_{\text{CDW}} \cdot \mathbf{f}_{\text{CDW}} + \delta^{3}\text{He}_{\text{ASW}} \cdot \mathbf{f}_{\text{ASW}} + 0 \cdot \mathbf{f}_{\text{GMW}} \tag{2}$$

Equation 2 is written to emphasize that the expectation is for $\delta^{3}\text{He} = 0$ in glacial melt, because the ${}^{3}\text{He}/{}^{4}\text{He}$ ratio is normalized to air, and air bubbles are the source of helium in glacial ice.

To capture the potential range of $\delta^{3}\text{He}$ that is brought on by variations in the water mass properties, the OMP solution and $\delta^{3}\text{He}$ reconstruction for PIB were randomly resampled using a Bootstrap method[62]. Appreciating that 2007 and 2014 were climatologically distinct years[21], we use the extrema in potential temperature, salinity, neon, and $\delta^{3}\text{He}$ from both years to define the water mass variability. The tracer values and uncertainties within each water mass are assumed to be normally distributed with parameters ($\mu$, $\sigma$), and listed in Supplementary Table 1. The OMP solution was resampled with 1500 iterations to define the parameter space (Fig. 2b, gray shading). The fit quality, quantified by $\|1 - r_i\|$, where $r_i$ are the model-data misfit, was better than 0.97 for all of the samples from 2007 and 2014, including those from PIB. Typically, a fit quality of 0.95 or better is considered acceptable level misfit[63].

**Geothermal heat flux calculation**. The bulk volcanic heat flux across the ice shelf front can be estimated by discretely integrating the scalar product of the convective mantle heat content (inferred from the ${}^{3}$He excess found in seawater) with the seawater velocity perpendicular ($u_{\text{perp},i}$) to the ice shelf,

$$Q = \sum_i \left[ {}^{3}\text{He}_{\text{exc}} \right] \cdot \text{HR} \cdot u_{\text{perp},i} \cdot \rho_{\text{sw}}\text{d}A \tag{3}$$

The $[{}^{3}\text{He}_{\text{exc}}]_i$ values have been interpolated onto the grid of perpendicular velocities at the ice shelf front and d$A$ is the surface area of each velocity grid cell (80 m${}^{2}$). The velocity data were obtained by the shipboard Acoustic Doppler Current Profiler (SADCP), during JR294 in a period coincident with the 2014 water sampling for helium and neon. The SADCP data from JR294 are broadly consistent with the established ice shelf cavity circulation; tides are weak in PIB, but the strength of meltwater outflows varies interannually by tens of percent[24]. The SADCP penetrates to 600 m, so the flow beneath 600 m is not resolved. However, the water column below 600 m is predominantly CDW and we found almost no glacial meltwater nor mantle-derived ${}^{3}$He below that depth.

**Estimation of uncertainty in the geothermal heat flux**. We identified three principal sources of uncertainty in the computation of $Q$ using Eq. (3): uncertainty in the ratio of ${}^{3}$He to mantle heat (HR), variations in the magnitude of water velocity at the cavity front ($u_{\text{perp}}$), and error introduced by interpolating coarse ${}^{3}$He measurements onto the finer resolution velocity field. Here we discuss these terms, respectively.

The calculation of heat content in J kg${}^{-1}$ seawater is based upon literature values of the ${}^{3}\text{He}/\text{HR}$[64]. For the present estimate, we use values of HR that fit the geologic description of the WAIS rift. The WAIS rift system is described as a composition of predominantly shield volcanoes, with no apparent plate motion underneath West Antarctica[47], and characteristics that are similar to certain island arc volcano systems, such as the Canary Island volcanoes[7]. Based upon these descriptions, estimates of HR from the literature were taken from regions where active vents are known to occur. The average HR from these measurements is 17 ± 6 × 10${}^{16}$ J per mol ${}^{3}$He and these studies are summarized in Supplementary Table 2.

Seawater flow within an ice shelf cavity is first order geostrophic, and can be reproduced using the thermal wind balance[65]. PIB follows this flow pattern, with persistent flow features including a strong meltwater jet on the west side of the ice shelf[33], partly as a result of weak tides in the Amundsen Sea[66]. Although the flow field is relatively stable, the strength of the cavity circulation and the magnitude of the velocity at the ice shelf front are known to vary by ca. 20% between years[21]. We use this uncertainty and the spatial mean of $u_p$ (0.03 ms${}^{-1}$) to estimate the uncertainty in the heat flux estimate that is introduced by seasonal to annual variations in the ice shelf cavity circulation (var[$u_{\text{perp}}$] = $(0.2\overline{u}_{\text{perp}})^{2}$).

It is apparent from the velocity field that the geochemical data do not capture all of the variations in the flow field (Fig. 4). For example, water samples for helium were not collected in the strong outflow observed near km 5, which has a high meltwater concentration[34]. To examine how the coarse resolution in ${}^{3}$He could affects the heat flux estimate from Eq. (3), both $\delta^{3}\text{He}$ and potential temperature were interpolated onto the SADCP grid. The total area, covered by the SADCP grid, which extends to a maximum of ~ 600 m (Fig. 4) is 25.25 km${}^{2}$. Potential temperature is 1 m vertical resolution, compared with ca. 75 m for ${}^{3}$He, and the two tracers broadly follows the same pattern (e.g., high in CDW; low in ASW and GMW). Therefore, potential temperature can serve as a higher resolution proxy for ${}^{3}$He. We used a linear regression ($R^2 = 0.79$, Supplementary Figure 1) to produce "proxy" ${}^{3}$He values from potential temperature. By comparing the spatial variations in the ${}^{3}$He from the mixing model (${}^{3}\text{He}_{\text{OMP}}$, described in Eq. (2)) and the proxy ${}^{3}$He (${}^{3}\text{He}_{\text{pr}}$) grids, we have a measure of the uncertainty produced by interpolating ${}^{3}$He from coarse to fine resolution. The spatial variations are captured by applying

the norm of the gradient operator, $\|\nabla\| = \left[()^2/dx^2 + ()^2/dy^2\right]^{0.5}$ to both grids and we computed the uncertainty in the coarsely sampled $^3$He grid (GErr) as the modulus of gradient of $^3$He$_{OMP}$ and the proxy $^3$He ($^3$He$_{pr}$, Supplementary Figure 1),

$$\text{GErr}(\%) = \frac{\left\|\nabla\,^3\text{He}_{OMP} - {}^3\text{He}_{pr}\right\|}{^3\text{He}_{OMP}}\frac{1}{\|\nabla\|} \times 100 \tag{4}$$

These are used, instead of the observed $^3$He, because the observed data contain the additional $^3$He source, which does not conform to the linear relationship between $^3$He and temperature. Depth profiles of the $^3$He measurements can be found in Supplementary Figure 2.

We find the coarse distribution of helium samples introduces up to 14% uncertainty into the full-scale $^3$He$_{exc}$ estimates (Supplementary Figure 4). In addition, the SD in literature HR values indicates additional uncertainty of 36% on the heat flux calculation (Supplementary Table 2, Eq. (3)) and the variability in the seawater velocity magnitude introduces another 20% uncertainty. These three sources of error are propagated using a Taylor expansion to express the variance in Eq. (3) for $Q$ in Watts,

$$\text{var}[Q] = \text{var}[{}^3\text{He}_{exc}]\left(\frac{dQ}{d\,^3\text{He}_{exc}}\right)^2 + \text{var}[\text{HR}]\left(\frac{dQ}{d\text{HR}}\right)^2 + \text{var}\left[u_{perp}\right]\left(\frac{dQ}{du_{perp}}\right)^2$$

$$\text{var}[Q] = \left(\frac{\text{GErr}}{100}\overline{^3\text{He}_{exc}}\right)^2\left(\sum_i \text{HR}u_{perp}\rho_{sw}dA\right)^2 \dots$$

$$+\text{var}[\text{HR}]\left(\sum_i {}^3\text{He}_{exc}u_{perp}\rho_{sw}dA\right)^2 \dots \tag{5}$$

$$+\left(0.2\overline{u}_{perp}\right)^2\left(\sum_i \text{HR}\,^3\text{He}_{exc}\rho_{sw}dA\right)^2$$

The term $\overline{^3\text{He}_{exc}}$ is the average of the excess $^3$He, which is computed as the difference between the observed $^3$He concentration and the reconstructed $^3$He from the OMP model. The term var[HR] is the variance in literature values listed in Supplementary Table 2, or $3.6 \times 10^{33}$ J mol$^{-1}$, and var[$u_{perp}$] is estimated as describe above.

**Data availability**. The $^3$He data presented in this study is available from the authors and from the Earthchem Library (http://www.earthchem.org/library/browse/view?id = 1152).

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

## Acknowledgements

We gratefully acknowledge the insights and comments of three anonymous reviewers. We thank the officers and crew of the RRS *James Clark Ross* and the RVIB *Nathaniel B. Palmer* for two excellent scientific expeditions in the Amundsen Sea. We thank Dempsey Lott III, Kevin Cahill, and Toby Koffman for analysis of the noble gas samples. This research was supported by the NSF Antarctic program through Award #1341630.

## Author contributions

K.H. and A.C.N.G. conceived the ocean2ice iSTAR A research plan in the Amundsen Sea. W.J.J. and P.S. carried out the noble gas analyses from samples collected in 2014 and 2007, respectively, at their individual facilities. D.V. provided glaciology and volcanology perspective to the analysis and interpretation. A.C.N.G. provided oceanographic perspective to the analysis and interpretation. B.L. lead the US participation in the UK iSTAR A project, developed the field sampling, and carried out analysis and interpretation.

## Additional information

**Competing interests:** The authors declare no competing interests.

