## [Peer Review File · Nature Communications]

Reviewers' comments:

Reviewer #1 (Remarks to the Author):

Review of "Evidence of an active volcanic heat source beneath the Pine Island Glacier" by Loose et al., for Nature Communications

The manuscript by Loose et al. uses new noble gas observations from the Amundsen Sea to deduce the presence of a volcanic heat source beneath the West Antarctic Ice Sheet, which could ultimately contribute to basal melting and influence ice-flow dynamics. The observations include two repeated section at the ending of the rapidly retreating Pine Island Ice Shelf, which are compared to a larger datasets of observations around Antarctica and in the Southern Ocean. The two sections near the Pine Island Ice Shelf show anomalously high ^3He content (a stable isotope derived from solid Earth emissions through volcanism), which cannot be explained by oceanic sources alone. The high ^3He anomaly is found in waters that, by their depth, temperature and neon signature, must contain a significant component of glacial meltwater, suggesting that the volcanic source is located beneath the WAIS. Because volcanic emissions of ^3He are generally accompanied by heat fluxes at relatively well-constrained ratios, the outflow of these ^3He -rich meltwaters can be converted to a heat flux, which appears to be of the same order of magnitude of active volcanoes or vent fields, suggesting substantial (albeit perhaps localized) volcanic activity under the WAIS. This could in turn alter basal melting and, speculatively, the flow dynamics of the ice sheet itself.

This is a nice piece of work that cleverly combines geochemical and oceanographic observations to deduce and quantify remote heat fluxes from under the West Antarctic Rift System. Because of my expertise, I can mostly comment on the novelty and the soundness of the noble gas, isotope, geochemical and oceanographic analysis. However, the implications for WAIS dynamics, or the novelty and importance of the results with respect to identifying and quantifying a heat source beneath the WARS fall outside my expertise. Other reviewers may be more qualified to comment on the broad implications of the work.

The dataset described represents a substantial new addition to existing He/Ne measurements in the Southern Ocean, and after perusing the figures and the text, I am convinced that the Authors' interpretation is sound, and the most logical explanation for the high ^3He anomalies near the Pine Island Glaciers has to be volcanic inputs carried by glacial meltwater. The use of a water mass mixing model constrained by T, S, Ne for the region is compelling, and the calculation of the heat flux from the hydrographic section satisfactory (although with some reservations- see below). Overall, the manuscript is generally well written and clear, the methods clearly laid out, and the figures informative. I have a few additional comments in the following.

- I think the Authors can have done a better job quantifying the importance of finding a localized source of heat underneath the WARS of the magnitude that they estimate, in the context of the other heat fluxes that may contribute to basal melting (e.g. heat diffusion from the crust, friction, etc.). This would have helped to convince that the deduced heat source is broadly important. For example, I would like to know how much basal meltwater can be explained by the volcanic that source, and how much by other sources (away from the influence of oceanic seawater beneath the ice-shelf).

- Also, the extent of the observations shown in the paper is a double-edged sword: while they allow to detect the volcanic source in meltwater at one specific location, they do not detect any other elsewhere, over a significant sector of the Antarctic coastline. Perhaps volcanic subglacial melting is not as relevant for the broad dynamics of the AIS? But as I noted above, these considerations may be better left to more expert reviewers.

- I think it is unlikely, but is there any way the ^3He and heat could be decoupled between the time

of their emission into basal meltwater and the time they reach the sea? Is there also the possibility of any contact with the atmosphere during this transit, or is the flow completely isolated? That is, is the total ^3He measured at the Pine Island Glacier section fully representative of the ^3He (and heat) emitted at the source?

- Line 81-83: There are observations in Fig. 2, between theta values of 0.5 and 1.0 C, that seem somewhat higher than the maxima cited in the text (10.2-10.6%). Are these from waters further north?

- Line 130-131. I am confused by the numbers: if the ^3He of the CDW end member is the same (9.15%), how can the values for 2014 and 2017, 12.2% and 12.3% (very close to each other), correspond to 25% and 35% excess compared to CDW? Table S1 only lists one property for CDW, although in the methods the Authors say that different end-member properties for different years were used. This should be clarified.

- Line 138-140, and model in equation (1). The Authors should discuss the possibility of biases in the velocity field from the ADCP instrument. The variability in ^3He observed between 2007 and 2014 could also be due to variations in the water velocity perpendicular to the section. There's the possibility that the transport across the section is variable and not in a steady-state. In that case the calculation with equation (1) would not be representative of the long term mean heat-flow carried by glacial meltwater. This may be an important source of uncertainty for the heat flux estimate, but it's hard to contain without repeated measurements of the velocity field. However, comparison with variability in velocity for different studies in similar settings may help supporting the robustness of the estimate.

- The mixing model could be more carefully validated. How well does it reconstruct water mass properties? Also, observed vs. predicted He_3 could be shown somewhere in the SI for completeness.

- Fig 2 and S1. Why is absolute Ne concentration shown rather than Ne saturation ($\Delta\text{-Ne}$)? I think the latter is more broadly used to discuss glacial sources, and perhaps more intuitive than the concentration, which can also be affected by temperature variations (albeit presumably small).

- Fig. 4. I am confused by the units in the first panel (MW). Since what is plotted is a profile with depth, shouldn't be the units expressed "per unit depth", e.g. MW/m? This way the total heat flux would be calculated by integration in the vertical.

- Fig. 4. Why are panel 2 and panel 3 separated?

- Lines 425-437. I struggled with this section (estimate of error from along-section interpolation). Could the Authors explain more clearly the rationale and result of the calculation? Also, in equation M3, what exactly is the norm of the operator in the denominator? Also, note that Fig. S3 seem to have incorrect units for $\Delta\text{-}^3\text{He}$.

- Table S2: I'm confused by the values in this table. Some of the "upper" values in column 3 seem to be smaller than the corresponding "lower" values in column 4, e.g. for the EPR13N, TAG26N, etc. Also, the "Average" value for the "lower" estimate, 6, does not seem to match the numbers in the column, which are all much larger.

- Abstract, line 29: add uncertainty range to the 2500MW heat flux estimate.

- Line 157: The figure 0.1278 has many digits - are all of them significant?

- Lines 157-162: for clarity the ^3H observations could also be converted to ^3He units, so that the

potential excess from 3H-derived ^3He could be readily compared with the other signals.

- Line 209: The Twitches Glacier could perhaps be shown on the map. Also, where would the outflow of this glacier to the sea be located? Maybe it also contributes excess ^3He to some other location.

- Line 221: remove "be" after "is not".

Reviewer #2 (Remarks to the Author):

Review of "Evidence for an active volcanic heat source..." by B. Loose et al.

This is an interesting paper and I recommend it for publication after some improvements. The authors make the case that water samples collected along the front of the Pine Island Glacier (PIG) have excess ^3He indicating the presence of a hydrothermal component. They then convert the excess ^3He to heat using $^3\text{He}/\text{heat}$ ratios published for submarine hydrothermal systems. This is a big leap, since $^3\text{He}/\text{heat}$ ratios vary by over an order of magnitude, and furthermore they have no idea what the proper value is for the supposed hydrothermal system under the PIG. They then proceed to combine acoustic Doppler profiler flow rates with their derived heat values to calculate a heat flux.

The paper is somewhat of a chore to read. The authors have left it up to the reader to decipher how the data correspond to the maps, etc. I have the following suggestions that would improve the paper:

1. It is difficult to read the maps in Figures 1 and 3. These figures should be much larger.
2. For Figure 4, where do the sections in the 2 right hand panels fall on the map? They should either include a small map with the figure, or explain where they fall. Or do these sections correspond to the data points shown in Figure 3? They should explain.
3. They should show some profiles of $\delta^3\text{He}$, neon, temperature, salinity, etc. vs. depth. This would help the reader understand what they are doing. It would not be necessary to show all the profiles, but some examples would be illustrative.
4. Were there any other chemical properties measured in these profiles besides ^3He and neon? For example, trace metals such as Fe and Mn would be clear indicators of hydrothermal input, and should be present in their profiles and should be correlated with $\delta^3\text{He}$. Trace metal anomalies would support their contention that this is hydrothermal ^3He .
5. Why do they need to use neon in order to estimate the % of glacial melt water? Salinity doesn't work? Or some other property? A standard theta-salt plot would be useful.
6. They should include a file of the actual ^3He , neon, depth data.

Some small points:

Line 64, they should write out Circumpolar Deep Water before reverting to the abbreviation. Not every reader may know this.

Line 212 messed up wording

Reviewer #3 (Remarks to the Author):

General Comments

This manuscript focuses on measurements of elevated Helium-3 to Helium-4 ratio in seawater

samples collected near the front of the Pine Island Glacier ice shelf. The authors interpret this regional helium isotope anomaly as evidence of subglacial volcanic activity which may be associated with the Hudson Mountain range. Empirical relationships are used by the authors to estimate the total heat flux associated with the purported subglacial volcanic system to be 2500 MW. They propose that understanding of subglacial rift volcanism in West Antarctica is needed to assess the future ice sheet stability.

There have been several papers suggesting current or recent subglacial volcanism in West Antarctica and highlighting its potential impact on ice sheet stability. Most of them are cited here but not Lough et al. (2013, Nat. Geo.), or Begeman et al. (2017, GRL) and Seroussi et al. (2017, JGR); the latter two having been just published. Because of new observations and modeling results the scientific interest in the potential impact of high geothermal flux and volcanic activity on the evolution of the West Antarctic ice sheet is rising. Hence, this study is timely and will be of interest to the broader scientific community. It takes the novel approach of using Helium isotope ratios in seawater collected in Amundsen and Ross Seas as a proxy of volcanic heat inputs beneath a major Antarctic outlet glacier experiencing rapid thinning (Pine Island Glacier). This work lies at the intersection of several disciplines, including isotope geochemistry, oceanography, volcanology, and glaciology. Its multidisciplinary nature should help draw attention of broad scientific audiences.

The work presented in this manuscript appears convincing. Admittedly I am not a specialist in Helium isotope geochemistry. The fact that their interpretation is based on finding a regionally anomalously high Helium isotope ratio is reassuring because it means that the core of their argument is based on relative magnitudes of the ratio rather than the ratio being above some threshold value determined by observations elsewhere. However, I would appreciate if the authors would discuss the possibility that the PIG Helium isotope anomaly may be caused by spatial variations in concentration of Helium-4 rather than Helium-3, as described here. It may be self explanatory to them but I would like to see an explanation for the reason for their focus on Helium-3 as the reason why the ratio varies spatially.

This paper will influence thinking in the field because it points the way to a novel approach towards detecting subglacial volcanism using isotopic and geochemical tracers along ice sheet margins. Direct access to subglacial zones will always be difficult and limited to a few places. Use of proxies of high heat flux around the ice sheet margin is highly promising. The study will likely inspire work in solid earth geophysics (e.g., seismology) which can help elucidate active volcanism and magmatism beneath the ice sheet (e.g., Lough et al., 2013). Ice sheet modelers can use the value of volcanic heat flux estimated here to verify the impact of this additional heat input on the behavior of Pine Island Glacier. Hence, I believe that this paper, if published, will have broad impact on different disciplines.

Specific Comments

Line 18 - 'convoluted topography' can be caused by many geologic processes. I presume that the authors mean subglacial volcanic features. This should be clarified. The critical aspect of subglacial volcanism that should be emphasized here is the fact that volcanic heat flux can be quite time-variable, as opposed to the background geothermal flux which, whether low or high, is relatively steady through time. This aspect should be emphasized here at least as much as volcanic topography. I even suggest dropping mention of the topography in favor of the temporal variability of heat flux. The topography itself seems tangentially important.

Line 45 - The complex rift system (WARS) that the authors are talking about stretches further out, under the Ross Ice Shelf and into the Ross Sea.

Line 46 - Volcanic activity along the WARS is an established fact (e.g., Mt. Erebus is a currently active volcano). Low elevation of a rifted continental crust is not inconsistent with volcanism. For instance the rifted continental crust in the Aegean Sea is well below sea level and has volcanoes

that have been active in Holocene (hence, would count as active volcanism under the common use of this term). I am not sure what the authors mean by 'the latest volcanic activity occurring ca. 25 Ma before present'. This is either a mis-statement or clearly not true since the Erebus volcano is active pretty much all the time at the present time and there are other volcanics younger than ca. 25 Ma in the WARS. This statement has to be removed or clarified. In addition, the authors should not make so many strong statements in a sentence without giving a citation, or citations, that support/s their statements.

Line 64 - Explain CDW the first time you use it.

Lines 76-77 - This is redundant. You already told us that Ra is the normalization factor in the equation given in Line 76.

Lines 204-208 - I continue to be puzzled why the well studied, young volcanic centers of Mt. Erebus and surroundings are being excluded here? What about all of the young volcanism in Marie Byrd Land that is proximal to Pine Island Glacier?

Line 214 - replace 'outflow' with 'trough'

Line 224 - Typically I discourage use of the word 'likely' unless it actually can be quantified in terms of likelihood. Is 1% or 10% or 90% probability of occurrence used here as 'likely'? Usually statements like this simply reflect authors' preference, not some concept of 'likelihood' in statistical meaning of this word. I recommend re-phrasing to 'preferred' or 'interpreted' but 'likely' should not be kept.

Lines 235 - 238 - I am missing a step here. The authors do not give enough information here to understand how they are getting the 0.02% fraction of glacial melt. Lets take a step back and first give the reader the total estimated volcanic heat flux and explain how this is estimated. Or is this estimate made on per-unit-volume of seawater basis (or maybe per-kg of seawater basis)? This critical step represents a major contribution of this manuscript and it should be explained clearly.

Lines 235 - 238 - I think that the 0.02% comparison is not the best one. Earth's climate system, including ocean heat content, is driven by the high density energy flux from solar irradiance. Average solar irradiance per unit area of Earth surface is almost 10,000 times greater than the average geothermal flux. Hence, geothermal flux, including any volcanic inputs, become only really important in settings where solar inputs are negligible (e.g., subglacial Antarctic environments). The heat used to melt PIG ice shelf is of solar origin and it is no surprise that it swamps any geothermal / volcanic inputs. A more logical comparison here would be between geothermal flux inputs (e.g., average continental geothermal flux times PIG subglacial area of about 175,000 km²) versus the volcanic input estimated here. Geothermal and volcanic inputs will help determine dynamics of the grounded ice, including its response to the ocean-driven melting along the coastal margins.

Lines 269-272 - I am not comfortable with the authors switching between volcanic heat source and mantle-derived heat source and treating the two as the same. After all the normal (conductive, non-volcanic) geothermal flux also is in a large part due to transport of heat from the mantle. So, it is 'mantle-derived' but this is not the type of heat source the authors mean. They talk about the heat flux that is advected to Earth surface (or near subsurface) by magma/lava. The authors should stick to just referring to 'volcanic heat flux' or something along these lines rather than switching between volcanic-, mantle-, hydrothermal- heat. It's confusing as it is presented right now. Begeman et al. (2017) is a very recent publication which provides a breakdown of different geologic processes that control geothermal flux magnitude and its spatial variability in Antarctica. It may be helpful in organizing the discussion in authors' manuscript:

Line 351 - I recommend using the terms: 'Inward' and 'Outward' rather than 'In' and 'Out'. It is

also unclear why the authors are not using MW for unit abbreviation here.

Line 405 and the paragraph starting with this line - This is another example of confusing changes in terminology used by the authors. Now their 'volcanic heat flux' is called 'geothermal heat flux'. This is not the case, geothermal flux cannot be estimated using Helium 3; it's also determined by heat conduction towards the Earth's surface. Helium 3 is not a tracer for conductive heat transport for Earth's interior. Pick one, most appropriate term for the type of heat that can be traced using Helium 3 and consistently use this one term.

Comments on the supplemental materials.

When you refer to the planet Earth, the word 'Earth' or 'Earth's' should be capitalized.

Table S1 - has the abbreviation GMW been explained anywhere?

References cited in this review

Lough, A. C., Wiens, D. A., Grace Barcheck, C., Anandakrishnan, S., Aster, R. C., Blankenship, D. D., Wilson, T. J. (2013). Seismic detection of an active subglacial magmatic complex in Marie Byrd Land, Antarctica. *Nature Geoscience*, 6(12), 1031–1035. <https://doi.org/10.1038/ngeo1992>

Begeman, C.B., Tulaczyk, S.M. and Fisher, A.T., 2017. Spatially variable geothermal heat flux in West Antarctica: evidence and implications. *Geophysical Research Letters*, 44(19), 9823-9832, doi 10.1002/2017GL075579

Seroussi, H., E. R. Ivins, D. A. Wiens, and J. Bondzio (2017), Influence of a West Antarctic mantle plume on ice sheet basal conditions, *J. Geophys. Res. Solid Earth*, 122, 7127–7155, doi:10.1002/2017JB014423.

Response to Reviewers:

We greatly appreciate the constructive and insightful comments from the three anonymous Reviewers. Their feedback has provided much-needed perspective on how to effectively communicate results from this study to a range of audiences. This was especially helpful where our study tended to intersect with concepts that are common to volcanology and glaciology, some of which needed to be fine-tuned.

We have responded to the respective comments from each of the Reviewers and have taken steps to change, clarify or fix the text, tables and figures. Please find our responses in line below.

Reviewer #1 (Remarks to the Author):

Review of "Evidence of an active volcanic heat source beneath the Pine Island Glacier" by Loose et al., for Nature Communications

The manuscript by Loose et al. uses new noble gas observations from the Amundsen Sea to deduce the presence of a volcanic heat source beneath the West Antarctic Ice Sheet, which could ultimately contribute to basal melting and influence ice-flow dynamics. The observations include two repeated sections at the ending of the rapidly retreating Pine Island Ice Shelf, which are compared to a larger dataset of observations around Antarctica and in the Southern Ocean. The two sections near the Pine Island Ice Shelf show anomalously high ^3He content (a stable isotope derived from solid Earth emissions through volcanism), which cannot be explained by oceanic sources alone. The high ^3He anomaly is found in waters that, by their depth, temperature and neon signature, must contain a significant component of glacial meltwater, suggesting that the volcanic source is located beneath the WAIS. Because volcanic emissions of ^3He are generally accompanied by heat fluxes at relatively well-constrained ratios, the outflow of these ^3He -rich meltwaters can be converted to a heat flux, which appears to be of the same order of magnitude of active volcanoes or vent fields, suggesting substantial (albeit perhaps localized) volcanic activity under the WAIS. This could in turn alter basal melting and, speculatively, the flow dynamics of the ice sheet itself.

This is a nice piece of work that cleverly combines geochemical and oceanographic observations to deduce and quantify remote heat fluxes from under the West Antarctic Rift System. Because of my expertise, I can mostly comment on the novelty and the soundness of the noble gas, isotope, geochemical and oceanographic analysis. However, the implications for WAIS dynamics, or the novelty and importance of the results with respect to identifying and quantifying a heat source beneath the WAIS fall outside my expertise. Other reviewers may be more qualified to comment on the broad implications of the work.

We thank the Reviewer for the affirmative feedback.

The dataset described represents a substantial new addition to existing He/Ne measurements in the Southern Ocean, and after perusing the figures and the text, I am convinced that the Authors' interpretation is sound, and the most logical explanation for the high ^3He anomalies near the Pine Island Glaciers has to be volcanic inputs carried by glacial meltwater. The use of a water mass mixing model constrained by T, S, Ne for the region is compelling, and the calculation of the heat flux from the hydrographic section satisfactory (although with some reservations- see below). Overall, the manuscript is generally well written and clear, the methods clearly laid out, and the figures informative. I have a few additional comments in the following.

- I think the Authors can have done a better job quantifying the importance of finding a localized source of heat underneath the WARS of the magnitude that they estimate, in the context of the other heat fluxes that may contribute to basal melting (e.g. heat diffusion from the crust, friction, etc.). This would have helped to convince that the deduced heat source is broadly important. For example, I would like to know how much basal meltwater can be explained by the volcanic that source, and how much by other sources (away from the influence of oceanic seawater beneath the ice-shelf).

We have added a section titled “Implications of such a volcanic heat source beneath Pine Island Glacier” near line 300 that describes previous estimates of the heat flux per unit area, which is the quantity most directly associated with melt. We compare with heat flux measurements taken from subsea vents and with a modeling study that recently simulated the effects of a mantle plume (Seroussi, et al., 2017).

Most measurements of heat flux at the ice sheet base are expressed in heat flux per unit area (e.g. Watts m^{-2}); these are often derived from a 1D flux parameterization, based on the geothermal gradient. In this case, we have a measure of the bulk heat flux, but we lack information on the size or extent of the hydrothermal features that source the 3He and heat to the atmosphere.

We are not able to convert the estimates of bulk volcanic heat into heat flux per unit area, which is the intensive property that would more directly reveal melting. This is because we don't have a constraint on the surface area of the volcanic heat release. We use the characteristic plume dimensions described by Seroussi et al. (2017) to postulate possible heat flux intensities, but it was not possible to compute the meltwater production definitively.

- Also, the extent of the observations shown in the paper is a double-edged sword: while they allow to detect the volcanic source in meltwater at one specific location, they do not detect any other elsewhere, over a significant sector of the Antarctic coastline. Perhaps volcanic subglacial melting is not as relevant for the broad dynamics of the AIS? But as I noted above, these considerations may be better left to more expert reviewers.

We take the Reviewer's point that the uniqueness of these helium-3 measurements might suggest that this mantle heat source is unique to the Pine Island Ice Shelf, rather than broadly characteristic of basal conditions beneath the WAIS. But the ability to detect mantle-driven melt at the front of the ice shelf is predicated on a well-defined hydrologic exchange beneath the base of the ice sheet and the coastal ocean. More common may be that basal meltwater may remain trapped beneath the ice sheet or found in subglacial lakes. The second Reviewer has directed us to a model study by Seroussi et al., (2017). They explore the effect on ice sheet melt rate that is produced by introducing a mantle plume in several regions of the WAIS. They find compelling evidence that mantle heat sources are part of the basal heat flux that occurs along the Whillans Ice Stream. Seroussi et al. did not include the Pine Island Ice Shelf within their model domain, but interestingly they found that the melt rates beneath the adjacent Thwaites glacier were not very sensitive to the introduction of a mantle plume. We include this in our discussion.

- I think it is unlikely, but is there any way the 3He and heat could be decoupled between the time of their emission into basal meltwater and the time they reach the sea? Is there also the possibility of any contact with the atmosphere during this transit, or is the flow completely isolated? That is, is the total 3He measured at the Pine Island Glacier section fully representative of the 3He (and heat) emitted at the source?

If the ^3He that we observe had already had a chance to exchange with the atmosphere, it would suggest that we are underestimating the mantle helium (and thus the mantle heat) associated with this source. However, the available evidence suggests that the helium-3 observed at the ice shelf front has not had the time to ventilate to the atmosphere. The samples were collected close to the ice shelf and many, but not all, samples showing high helium-3 were found in stratified waters that were not in contact with the atmosphere. Upstream of the ice shelf, the ice sheet is quite thick, suggesting no opportunity for air equilibration.

We have included further discussion about how the $3\text{He}/4\text{He}$ isotope ratio might be altered in the section entitled "What are the possible sources of 3He production?", starting at line 146.

- Line 81-83: There are observations in Fig. 2, between theta values of 0.5 and 1.0 C, that seem somewhat higher than the maxima cited in the text (10.2-10.6%). Are these from waters further north?

Thank you for pointing this out, we re-examined each of the data sets included in Figure 2, and it appears we mis-reported the maximum value from the Pacific sector, which should be 10.9. We apologize for mis-reporting these data. The increase in the offshore value for the Pacific sector does not alter our interpretation of the results. We have modified Line 90 to reflect this change.

- Line 130-131. I am confused by the numbers: if the 3He of the CDW end member is the same (9.15%), how can the values for 2014 and 2017, 12.2% and 12.3% (very close to each other), correspond to 25% and 35% excess compared to CDW? Table S1 only lists one property for CDW, although in the methods the Authors say that different end-member properties for different years were used. This should be clarified.

We used the variation in the core CDW value between 2007 and 2014 to define the uncertainty that is expressed in Table S1. However, we take the Reviewer's point that the sentence implies that two different end members were used. We have modified the sentence as follows: "A fork in the $d^3\text{He}$ distribution occurred between $q = -1.5$ and -0.5 °C, with $d^3\text{He}$ exceeding the average CDW end member (9.15 ± 0.65) in 2014 by up to 33% ($d^3\text{He} = 12.2$) and in 2007 by 35% ($d^3\text{He} = 12.3$)"

- Line 138-140, and model in equation (1). The Authors should discuss the possibility of biases in the velocity field from the ADCP instrument. The variability in 3He observed between 2007 and 2014 could also be due to variations in the water velocity perpendicular to the section. There's the possibility that the transport across the section is variable and not in a steady-state. In that case the calculation with equation (1) would not be representative of the long term mean heat-flow carried by glacial meltwater. This may be an important source of uncertainty for the heat flux estimate, but it's hard to contain without repeated measurements of the velocity field. However, comparison with variability in velocity for different studies in similar settings may help supporting the robustness of the estimate.

In the first version of this manuscript, we identified interpolation errors as potentially the most important source of error in the heat flux calculation. However, we take the Reviewer's point about how stable the 'snapshot' of the magnitude of the velocity field is, with regard to the long and short-term variations in the cavity flow velocity. We have included a discussion of the variations in cavity circulation and estimates of how those affect the velocity field, and we have acknowledged this effect in the uncertainty on heat flow. This new discussion is found near lines 472-483: "Seawater flow within an ice shelf cavity is first order geostrophic, and can be well reproduced using the thermal wind balance⁶¹. Pine Island Bay follows this flow pattern, with persistent flow features including a strong meltwater jet on the west side of the ice shelf²⁹, partly as a result of weak tides in the Amundsen Sea⁶². While the flow field is relatively stable, the

strength of the cavity circulation and the magnitude of the velocity at the ice shelf front are known to vary by ca. 20% between years¹⁷. We use this uncertainty and the spatial mean of u_p (0.03 ms^{-1}) to estimate the uncertainty in the heat flux estimate that is introduced by seasonal to annual variations in the ice shelf cavity circulation”.

While the magnitude of the cavity circulation is known to vary, the overall structure of the flow follows the thermal wind balance and is very weakly tidal, so it seems less likely that changes in the structure and location of the inflows/outflows would be responsible for the differences we observed between 2007 and 2014.

- The mixing model could be more carefully validated. How well does it reconstruct water mass properties? Also, observed vs. predicted He3 could be shown somewhere in the SI for completeness.

We have added a statement to the methods section M1 that describes the fit quality of the mixing model “The fit quality, quantified by $\|1 - r_i\|$, where r_i are the model-data misfit, was better than 0.97 for all of the samples from 2007 and 2014, including those from Pine Island Bay. Typically a fit quality of 0.95 or better is considered acceptable level misfit⁶⁴.” Figures 2 and S1 both show the observed ^3He (colored circles) and the range of predicted ^3He from the mixing model.

- Fig 2 and S1. Why is absolute Ne concentration shown rather than Ne saturation (ΔNe)? I think the latter is more broadly used to discuss glacial sources, and perhaps more intuitive than the concentration, which can also be affected by temperature variations (albeit presumably small).

We find examples in the literature of both $[\text{Ne}]$ concentration (e.g. Schlosser et al., and ΔNe to visualize the presence of meltwater. As the reviewer indicates, the saturation anomaly is affected by water temperature and salinity, so that ΔNe can reflect the combined effects of these tracers, which somewhat complicates the interpretations, so we have used the $[\text{Ne}]$ concentration to depict meltwater distributions, which are apparent, even when viewing the concentration values. We have included a plot, as requested, of ΔNe in the supplemental materials.

- Fig. 4. I am confused by the units in the first panel (MW). Since what is plotted is a profile with depth, shouldn't be the units expressed “per unit depth”, e.g. MW/m? This way the total heat flux would be calculated by integration in the vertical.

The 2D grid of heat flux estimates has a spatial resolution of $dx = 100$, $dy = 8$ m. We have explicitly included those terms in the discrete row-wise sum of equation (1) over the grid. This produces a column of heat flux values in MW at each vertical point in the discrete grid. We have rephrased the figure caption in the hope that this will clarify to readers who may have the same question the Reviewer has posed: “Left panel shows the discrete sum of Inward, Outward, and Net mantle heat flux in Megawatts (MW) at each depth horizon in the heat flux grid.”

- Fig. 4. Why are panel 2 and panel 3 separated?

Panels 2 and 3 are separated because the ship transited those sections at different times and with different course headings, separated by approx. 24 hours. The separation alerts the reader that the measurements were discontinuous, and is consistent with Garabato et al., (2017) who presented the same data. We added this sentence to the legend: “The break at 32 km exists because the two sections were collected at different times, separated by less than 24 hours.”

- Lines 425-437. I struggled with this section (estimate of error from along-section interpolation). Could the Authors explain more clearly the rationale and result of the calculation? Also, in equation M3, what exactly is the norm of the operator in the denominator? Also, note that Fig. S3 seem to have incorrect units for Delta-3He.

Thank you for pointing out the error in Figure S3, we have fixed the caption.

We have revised several parts of section M2 to improve the explanation of the uncertainty estimate. This includes a revision of the text used to describe the uncertainty introduced by grid interpolation, and an improved estimate of the uncertainty to include effects of seasonal to annual variations in the flow field. Please find those changes in section M2.

- Table S2: I'm confused by the values in tis table. Some of the "upper" values in column 3 seem to be smaller than the corresponding "lower" values in column 4, e.g. for the EPR13N, TAG26N, etc. Also, the "Average" value for the "lower" estimate, 6, does not seem to match the numbers in the column, which are all much larger.

Thank you for pointing this out, we have exchanged the values to be consistent with the lower and upper limits. The "average" values at the bottom of the table was actually the average and the standard deviation. We have modified the table to more accurately depict what each of those two values represent.

- Abstract, line 29: add uncertainty range to the 2500MW heat flux estimate.

Uncertainty was added.

- Line 157: The figure 0.1278 has many digits - are all of them significant?

We have revised this number to 0.13.

- Lines 157-162: for clarity the 3H observations could also be converted to 3He units, so that the potential excess from 3H-derived 3He could be readily compared with the other signals.

We have included a conversion from d3He in % to d3He in moles/kg for comparison purposes: "For comparison purposes, $\delta^3\text{He} = 1$ corresponds to roughly 1.3×10^{-15} moles $^3\text{He kg}^{-1}$, or a factor of 100 greater than the tritiogenic ^3He "

- Line 209: The Twitches Glacier could perhaps be shown on the map. Also, where would the outflow of this glacier to the sea be located? Maybe it also contributes excess 3He to some other location.

We have added a label for the Thwaites Glacier to Figure 1.

- Line 221: remove "be" after "is not".

Thank you, we have removed.

Reviewer #2 (Remarks to the Author):

Review of "Evidence for an active volcanic heat source..." by B. Loose et al.

This is an interesting paper and I recommend it for publication after some improvements. The authors make the case that water samples collected along the front of the Pine Island Glacier (PIG) have excess ^3He indicating the presence of a hydrothermal component. They then convert the excess ^3He to heat using $^3\text{He}/\text{heat}$ ratios published for submarine hydrothermal systems. This is a big leap, since $^3\text{He}/\text{heat}$ ratios vary by over an order of magnitude, and furthermore they have no idea what the proper value is for the supposed hydrothermal system under the PIG. They then proceed to combine acoustic Doppler profiler flow rates with their derived heat values to calculate a heat flux.

The paper is somewhat of a chore to read. The authors have left it up to the reader to decipher how the data correspond to the maps, etc. I have the following suggestions that would improve the paper:

1. It is difficult to read the maps in Figures 1 and 3. These figures should be much larger.

We have increased the font size in Figures 1 and 3, and will request that they be typeset in large format.

2. For Figure 4, where do the sections in the 2 right hand panels fall on the map? They should either include a small map with the figure, or explain where they fall. Or do these sections correspond to the data points shown in Figure 3? They should explain.

Figure 3 shows a top view of the section in Figure 4. We have included the following sentence in the Figure 4 caption to reinforce that point: "Section along the edge of the Pine Island Ice Shelf in 2014 (as indicated by colored squares in Figure 3, right panel)."

3. They should show some profiles of $\delta^3\text{He}$, neon, temperature, salinity, etc. vs. depth. This would help the reader understand what they are doing. It would not be necessary to show all the profiles, but some examples would be illustrative.

We have added a new figure to the supplemental materials (Figure S1) that displays the profiles of ^3He , T, S, and DNe (the saturation anomaly of neon) for all the samples collected in 2007 and 2014.

4. Were there any other chemical properties measured in these profiles besides ^3He and neon? For example, trace metals such as Fe and Mn would be clear indicators of hydrothermal input, and should be present in their profiles and should be correlated with $\delta^3\text{He}$. Trace metal anomalies would support their contention that this is hydrothermal ^3He .

We did hope to detect glacial meltwater, but unfortunately the JCR is not able to be set up for trace metal clean sampling.

5. Why do they need to use neon in order to estimate the % of glacial melt water? Salinity doesn't work? Or some other property? A standard theta-salt plot would be useful.

Neon, in combination with temperature and salinity, provide the most accurate estimation of glacial melt water (Jenkins & Jacobs, 2008).

6. They should include a file of the actual ^3He , neon, depth data.

We proposed to make the data available in the public domain, through the CLIVAR CCHDO clearing house in accordance with NSF public data policy. This mechanism provides a coherent structure to ensure that relevant metadata and other accompanying materials are cross-referenced; these are the strengths of an ocean data center. We will ensure that the URL, DOI and other pertinent data set can be found in the acknowledgements.

Some small points:

Line 64, they should write out Circumpolar Deep Water before reverting to the abbreviation. Not every reader may know this.

Thank you for pointing this out, we intended to define it the first time it was used. The definition was included.

Line 212 messed up wording

Fixed, thank you.

Reviewer #3 (Remarks to the Author):

General Comments

This manuscript focuses on measurements of elevated Helium-3 to Helium-4 ratio in seawater samples collected near the front of the Pine Island Glacier ice shelf. The authors interpret this regional helium isotope anomaly as evidence of subglacial volcanic activity which may be associated with the Hudson Mountain range. Empirical relationships are used by the authors to estimate the total heat flux associated with the purported subglacial volcanic system to be 2500 MW. They propose that understanding of subglacial rift volcanism in West Antarctica is needed to assess the future ice sheet stability.

There have been several papers suggesting current or recent subglacial volcanism in West Antarctica and highlighting its potential impact on ice sheet stability. Most of them are cited here but not Lough et al. (2013, Nat. Geo.), or Begeman et al. (2017, GRL) and Seroussi et al. (2017, JGR); the latter two having been just published. Because of new observations and modeling results the scientific interest in the potential impact of high geothermal flux and volcanic activity on the evolution of the West Antarctic ice sheet is rising. Hence, this study is timely and will be of interest to the broader scientific community. It takes the novel approach of using Helium isotope ratios in seawater collected in Amundsen and Ross Seas as a proxy of volcanic heat inputs beneath a major Antarctic outlet glacier experiencing rapid thinning (Pine Island Glacier). This work lies at the intersection of several disciplines, including isotope geochemistry, oceanography, volcanology, and glaciology. Its multidisciplinary nature should help draw attention of broad scientific audiences.

We thank the Reviewer for pointing out these three studies on the topic that we had previously overlooked. We have included these studies in the interpretation of our results, and these citations can be found near lines 52, and 300.

The work presented in this manuscript appears convincing. Admittedly I am not a specialist in Helium isotope geochemistry. The fact that their interpretation is based on finding a regionally anomalously high Helium isotope ratio is reassuring because it means that the core of their argument is based on relative magnitudes of the ratio rather than the ratio being above some threshold value determined by

observations elsewhere. However, I would appreciate if the authors would discuss the possibility that the PIG Helium isotope anomaly may be caused by spatial variations in concentration of Helium-4 rather than Helium-3, as described here. It may be self explanatory to them but I would like to see an explanation for the reason for their focus on Helium-3 as the reason why the ratio varies spatially.

We take the Reviewer's point that discussion of how ^4He affects the $\delta^3\text{He}$ isotope ratio. We have added the following paragraph near Line 173 to clarify how ^4He might vary: "The $^3\text{He}/^4\text{He}$ isotope ratio that is used to compute $\delta^3\text{He}$ can also be affected by the production of ^4He through the decay chain that begins with ^{238}U , which is naturally present in many rock types within the continental crust, which consequently can make its way into groundwater³⁷. The $\delta^3\text{He}$ signal that we observe at the front of the Pine Island Ice Shelf, may include additional ^4He from crustal rocks, but this incorporation drives the $^3\text{He}/^4\text{He}$ isotope ratio in the opposite direction from that of the mantle, so additional ^4He production would result in a masking or underestimate of the total mantle helium. There are no known processes for removing ^4He gas, save bubble formation or diffusive degassing, which would affect all the dissolved gases in a similar manner."

This paper will influence thinking in the field because it points the way to a novel approach towards detecting subglacial volcanism using isotopic and geochemical tracers along ice sheet margins. Direct access to subglacial zones will always be difficult and limited to a few places. Use of proxies of high heat flux around the ice sheet margin is highly promising. The study will likely inspire work in solid earth geophysics (e.g., seismology) which can help elucidate active volcanism and magmatism beneath the ice sheet (e.g., Lough et al., 2013). Ice sheet modelers can use the value of volcanic heat flux estimated here to verify the impact of this additional heat input on the behavior of Pine Island Glacier. Hence, I believe that this paper, if published, will have broad impact on different disciplines.

Specific Comments

Line 18 - 'convoluted topography' can be caused by many geologic processes. I presume that the authors mean subglacial volcanic features. This should be clarified. The critical aspect of subglacial volcanism that should be emphasized here is the fact that volcanic heat flux can be quite time-variable, as opposed to the background geothermal flux which, whether low or high, is relatively steady through time. This aspect should be emphasized here at least as much as volcanic topography. I even suggest dropping mention of the topography in favor of the temporal variability of heat flux. The topography itself seems tangentially important.

We have reworded the first two sentences of the abstract based on Reviewer feedback, including the emphasis of the time-variable nature in volcanic heat: "Tectonic landforms and evidence of subglacial water flow indicate that volcanism may produce glacial melt beneath the West Antarctic Ice Sheet (WAIS). However, the ice sheet and the transient nature of volcanism make it extremely challenging to identify and to measure the associated heat flux. "

Line 45 - The complex rift system (WARS) that the authors are talking about stretches further out, under the Ross Ice Shelf and into the Ross Sea.

Thank you for clarifying this point about the geology, we have modified this sentence accordingly.

Line 46 - Volcanic activity along the WARS is an established fact (e.g., Mt. Erebus is a currently active volcano). Low elevation of a rifted continental crust is not inconsistent with volcanism. For instance the

rifted continental crust in the Aegean Sea is well below sea level and has volcanoes that have been active in Holocene (hence, would count as active volcanism under the common use of this term). I am not sure what the authors mean by 'the latest volcanic activity occurring ca. 25 Ma before present'. This is either a mis-statement or clearly not true since the Erebus volcano is active pretty much all the time at the present time and there are other volcanics younger than ca. 25 Ma in the WARS. This statement has to be removed or clarified. In addition, the authors should not make so many strong statements in a sentence without giving a citation, or citations, that support/s their statements.

We thank the Reviewer for pointing out these apparent mis-statements in our introduction to West Antarctic volcanism. We should have drawn a stronger distinction between the exposed volcanos (e.g. Erebus) and the subglacial volcanos, and been careful to mention the clear evidence of activity. We have rewritten this section as follows: "Determining subglacial heat flow beneath the WAIS is complicated by the presence an extensional volcanic rift system that stretches across Marie Byrd Land from the Pine Island glacier to the Ross Ice Shelf and into the Ross Sea^{7,8}. This is known as the West Antarctic Rift System (WARS). To date as many as 138 volcanoes have been identified throughout West Antarctica⁹, including the presently active Mt. Erebus¹⁰ along the Terror Rift, as well as Mt. Siple¹⁰, and Mt. Waesche¹¹ that both show evidence of recent activity. However, the locations and extent of volcanic activity along WARS is debated, because many of the 138 known volcano-like features are buried beneath several km of ice, and there is evidence suggesting much of the interior subglacial WARS may be dormant^{12,13}. Yet, recent direct measurement of the thermal gradient beneath the Whillans Ice Stream reveal heat flux that far exceeds the background geothermal gradient⁴, and contradict the assertion that the interior WARS is dormant. The apparent surface deformations in the thickness of the WAIS also suggest localized heat fluxes that are most likely volcanic due to their intensity^{14,15}, and ash layers from ice cores reveal more recent eruptions¹⁶. The detection of earthquakes as recently as 2010, suggest magma migration beneath the Executive Committee mountains, in a region of Marie Byrd Land where seismic studies have revealed thin crust and low density mantle material beneath^{13"}.

Line 64 - Explain CDW the first time you use it.

Thank you for pointing this out, we intended to define it the first time it was used. The definition is included.

Lines 76-77 - This is redundant. You already told us that Ra is the normalization factor in the equation given in Line 76.

We have removed the redundancy.

Lines 204-208 - I continue to be puzzled why the well studied, young volcanic centers of Mt. Erebus and surroundings are being excluded here? What about all of the young volcanism in Marie Byrd Land that is proximal to Pine Island Glacier?

This section is focused on the mostly subglacial volcanoes and those within the catchment of the Pine Island Glacier. We have modified the Introduction to correct previously noted omission of Mt. Erebus.

Line 214 - replace 'outflow' with 'trough'

Changed, thank you.

Line 224 - Typically I discourage use of the word 'likely' unless it actually can be quantified in terms of likelihood. Is 1% or 10% or 90% probability of occurrence used here as 'likely'? Usually statements like this simply reflect authors' preference, not some concept of 'likelihood' in statistical meaning of this word. I recommend re-phrasing to 'preferred' or 'interpreted' but 'likely' should not be kept.

We substituted plausible for likely.

Lines 235 - 238 - I am missing a step here. The authors do not give enough information here to understand how they are getting the 0.02% fraction of glacial melt. Lets take a step back and first give the reader the total estimated volcanic heat flux and explain how this is estimated. Or is this estimate made on per-unit-volume of seawater basis (or maybe per-kg of seawater basis)? This critical step represents a major contribution of this manuscript and it should be explained clearly.

We have removed the comparison of 0.02%, as the Reviewer suggests that it does not serve as a helpful comparison, and added the following sentence to explain the computation: "We compute the $^3\text{He}_{\text{exc}}$ as the difference between the measured ^3He and the value predicted by the linear mixing model. The $^3\text{He}_{\text{exc}}$, expressed in mols kg^{-1} of seawater divided by HR provides an estimate of volcanic heat conten in Joules per kilogram of seawater (J kg^{-1})."

Lines 235 - 238 - I think that the 0.02% comparison is not the best one. Earth's climate system, including ocean heat content, is driven by the high density energy flux from solar irradiance. Average solar irradiance per unit area of Earth surface is almost 10,000 times greater than the average geothermal flux. Hence, geothermal flux, including any volcanic inputs, become only really important in settings where solar inputs are negligible (e.g., subglacial Antarctic environments). The heat used to melt PIG ice shelf is of solar origin and it is no surprise that it swamps any geothermal / volcanic inputs. A more logical comparison here would be between geothermal flux inputs (e.g., average continental geothermal flux times PIG subglacial area of about 175,000 km^2) versus the volcanic input estimated here. Geothermal and volcanic inputs will help determine dynamics of the grounded ice, including its response to the ocean-driven melting along the coastal margins.

We reworded this section, as follows: "The excess neon found in samples with excess ^3He reveals a connection between mantle helium and glacial meltwater production, which is consistent with volcanic heat producing subglacial melt beneath the grounded Pine Island Glacier. We have estimated this volcanic heat content using the average of 17 reported estimates of ^3He /heat ratio (HR) from subsea hydrothermal vents. Lupton et al.,⁴⁹ provide a summary of the HR values, while Jenkins et al.,³⁸ give a recent estimate for the Atlantic spreading center. The mean and standard deviation of the literature values from subsea floor vents yield a ^3He /heat ratio of $\text{HR} = 17 \pm 6 \times 10^{16}$ Joules per mol ^3He (Table S2). We compute the ^3He excess ($^3\text{He}_{\text{exc}}$) as the difference between the measured ^3He and the value predicted by the linear mixing model. The $^3\text{He}_{\text{exc}}$, expressed in mols kg^{-1} of seawater divided by HR provides an estimate of volcanic heat conten in Joules per kilogram of seawater (J kg^{-1}). Based on the observed ^3He excesses, the mantle-derived heat at the front of the ice shelf cavity is $32 \pm 12 \text{ J kg}^{-1}$ of seawater. This excess heat is small compared to the heat content of CDW²⁰ (ca. 12 kJ kg^{-1}), demonstrating that volcanic heat does not contribute significantly to the glacial melt observed in the ocean at the front of the ice shelf. This is consistent with our understanding of melt dynamics beneath the Pine Island Ice Shelf – that most of the basal melt occurs within the cavity²⁰. Yet, this relatively dilute heat source may be much more concentrated at the time of contact with the ice sheet, and the magnitude more significant when compared to the background geothermal heat supply to the grounded glacier. We

can infer the heat flux to the ice sheet using observations of seawater velocity and our understanding of the ice shelf cavity circulation.”

Lines 269-272 - I am not comfortable with the authors switching between volcanic heat source and mantle-derived heat source and treating the two as the same. After all the normal (conductive, non-volcanic) geothermal flux also is in a large part due to transport of heat from the mantle. So, it is ‘mantle-derived’ but this is not the type of heat source the authors mean. They talk about the heat flux that is advected to Earth surface (or near subsurface) by magma/lava. The authors should stick to just referring to ‘volcanic heat flux’ or something along these lines rather than switching between volcanic-, mantle-, hydrothermal- heat. It’s confusing as it is presented right now. Begeman et al. (2017) is a very recent publication which provides a breakdown of different geologic processes that control geothermal flux magnitude and its spatial variability in Antarctica. It may be helpful in organizing the discussion in authors’ manuscript:

We have struggled with the terminology that strikes the best balance. The process transferring helium-3 into the subglacial hydrologic system is contact between hydrothermal fluids and magmatic rock or more likely magmatic gases. However, it is cumbersome to continually refer to the magmatic-hydrothermal heat flux. Much of the literature refers to subglacial volcanism as a catch-all term, which we could adopt, but the literature on ^3He refers to the mantle as this is the ultimate source of helium-3. Based on the Reviewer’s suggestion for consistency, we have added the following sentence at line 326: “Mantle ^3He escapes during magma degassing, which produces steam and volatile gas transport in adjacent hydrothermal fluids ⁴². Even if the glacial debris is rich in basalts, these cooled magmas have already lost much of their ^3He burden during the cooling process. Hereafter, we refer to the magma-driven hydrothermal heat transport as the ‘volcanic heat flux’.”

We have tried to restrict the use of the term ‘magma’ and ‘hydrothermal’ to only discussion of citations that uses the same terms.

Line 351 - I recommend using the terms: ‘Inward’ and ‘Outward’ rather than ‘In’ and ‘Out’. It is also unclear why the authors are not using MW for unit abbreviation here.

We have changed them to Inward and Outward and include (MW) in parentheses to affirm to the reader the meaning of the abbreviation.

Line 405 and the paragraph starting with this line - This is another example of confusing changes in terminology used by the authors. Now their ‘volcanic heat flux’ is called ‘geothermal heat flux’. This is not the case, geothermal flux cannot be estimated using Helium 3; it’s also determined by heat conduction towards the Earth’s surface. Helium 3 is not a tracer for conductive heat transport for Earth’s interior. Pick one, most appropriate term for the type of heat that can be traced using Helium 3 and consistently use this one term.

Please see response to above comment on Lines 269-272.

Comments on the supplemental materials.

When you refer to the planet Earth, the word ‘Earth’ or ‘Earth’s’ should be capitalized.

We have capitalized Earth.

Table S1 - has the abbreviation GMW been explained anywhere?

We have added a complete caption to Table S1 including the definition of GMW.

References cited in this review

Lough, A. C., Wiens, D. A., Grace Barcheck, C., Anandakrishnan, S., Aster, R. C., Blankenship, D. D., Wilson, T. J. (2013). Seismic detection of an active subglacial magmatic complex in Marie Byrd Land, Antarctica. *Nature Geoscience*, 6(12), 1031–1035. <https://doi.org/10.1038/ngeo1992>

Begeman, C.B., Tulaczyk, S.M. and Fisher, A.T., 2017. Spatially variable geothermal heat flux in West Antarctica: evidence and implications. *Geophysical Research Letters*, 44(19), 9823–9832, doi 10.1002/2017GL075579

Seroussi, H., E. R. Ivins, D. A. Wiens, and J. Bondzio (2017), Influence of a West Antarctic mantle plume on ice sheet basal conditions, *J. Geophys. Res. Solid Earth*, 122, 7127–7155, doi:10.1002/2017JB014423.

REVIEWERS' COMMENTS:

Reviewer #1 (Remarks to the Author):

This is my second review of the manuscript by Loose et al., and it refers to the revised version. A careful re-reading of the manuscript indicates that the Authors have taken my and other reviewers' suggestions seriously, and as a result clarified my previous concerns. As identified in my previous review, I think the data and analysis provide credible evidence for sub-glacial volcanism; other Reviewers have pointed out the significance of the finding for geology/glaciology, and the interest it may raise for the broader community. I am therefore supportive of publication.

A few remaining things:

- Abstract, line 29: remove "in" after "comparable"
- Line 315: "would imply": it is not clear what the subject of this sentence is. Maybe add "it"?
- Lines 368-369, "circular logic": maybe there's a better way to convey the point? "Circular logic" doesn't seem super clear.
- Line 484: change "course" to "coarse"

Reviewer #3 (Remarks to the Author):

I reviewed the revised manuscript, supplemental materials, and the response to reviewers. The authors have done a great job at responding to reviewers' comments. I have not found anything to comment in the new version of the manuscript except two small things:

- The end of the first full paragraph on Page 2 -> the word 'unstable' should be used instead of 'instable'. The latter is a French word for 'unstable'.
- There are missing https addresses at the end of the acknowledgement section.

REVIEWERS' COMMENTS: Reviewer #1 (Remarks to the Author):

This is my second review of the manuscript by Loose et al., and it refers to the revised version. A careful re-reading of the manuscript indicates that the Authors have taken my and other reviewers' suggestions seriously, and as a result clarified my previous concerns. As identified in my previous review, I think the data and analysis provide credible evidence for sub-glacial volcanism; other Reviewers have pointed out the significance of the finding for geology/glaciology, and the interest it may raise for the broader community. I am therefore supportive of publication.

A few remaining things:

- Abstract, line 29: remove "in" after "comparable" □

- Line 315: "would imply": it is not clear what the subject of this sentence is. Maybe add "it"? □-

We have fixed this as part of reducing the length of the abstract.

Lines 368-369, "circular logic": maybe there's a better way to convey the point? "Circular logic" doesn't seem super clear.

We have attempted to rephrase this statement as follows: "Simulations of the adjacent Thwaites Glacier may suggest that such a heat source will not significantly alter the subglacial melt rate in comparison with the high rate of friction⁵⁸, but this could be circular argument if volcanic heat supply is already part of the recipe of processes leading to high velocity and frictional heating of the ice streams in the Pine Island and Thwaites Glacier."

- Line 484: change "course" to "coarse"

Fixed, thank you.

Reviewer #3 (Remarks to the Author):

I reviewed the revised manuscript, supplemental materials, and the response to reviewers. The authors have done a great job at responding to reviewers' comments. I have not found anything to comment in the new version of the manuscript except two small things:

- The end of the first full paragraph on Page 2 -> the word 'unstable' should be used instead of 'instable'. The latter is a French word for 'unstable'.

Fixed, thank you.

- There are missing https addresses at the end of the acknowledgement section.

We have added that URL now that the data is published under a DOI.